# Anti-Coking and Anti-Sintering Ni/Al₂O₃ Catalysts in the Dry Reforming of Methane: Recent Progress and Prospects

Xingyuan Gao [1,2,†], Zhiyong Ge [1,†], Guofeng Zhu [1], Ziyi Wang [1], Jangam Ashok [3] and Sibudjing Kawi [3,*]

1    Department of Chemistry and Material Science, Guangdong University of Education,
     Guangzhou 510303, China; gaoxingyuan@gdei.edu.cn (X.G.); gezhiyong@gdei.edu.cn (Z.G.);
     zhuguofeng@gdei.edu.cn (G.Z.); wangzy@gdei.edu.cn (Z.W.)
2    Engineering Technology Development Center of Advanced Materials and Energy Saving and Emission
     Reduction in Guangdong Colleges and Universities, Guangzhou 510303, China
3    Department of Chemical and Biomolecular Engineering, National University of Singapore,
     Singapore 117585, Singapore; chejang@nus.edu.sg
*    Correspondence: chekawis@nus.edu.sg; Tel.: +65-6516-6312
†    The authors contribute equally to this review.

**Abstract:** Coking and metal sintering are limitations of large-scale applications of Ni/Al₂O₃ catalysts in DRM reactions. In this review, several modification strategies to enhance the anti-deactivation property of Ni/Al₂O₃ are proposed and discussed with the recently developed catalyst systems, including structure and morphology control, surface acidity/basicity, interfacial engineering and oxygen defects. In addition, the structure–performance relationship and deactivation/anti-deactivation mechanisms are illustrated in depth, followed by prospects for future work.

**Keywords:** Ni; Al₂O₃; dry reforming of methane (DRM); anti-coking; anti-sintering





## 1. Introduction

In recent years, with the sustainable industrial development of human society and continuous growth of the population, the further exploitation of limited natural resources such as fossil fuels has attracted increasing attention [1]. The increasing energy consumption has led to the global greenhouse effect, which has caused great damage to the world environment and ecology [1,2]. Therefore, in order to protect the environment and save resources, it is urgent to explore technologies and methods that can replace energy and make better use of natural gas and other fossil fuel derivatives [3]. Methane (CH₄) and carbon dioxide (CO₂) are two main greenhouse gases. It is of great significance to realize their comprehensive transformation and utilization, in order to slow down the greenhouse effect [4]. In comparison with the partial oxidation of methane (POM) and steam reforming of methane (SRM), dry reforming of methane (DRM) is preferred because it can convert two greenhouse gases (CH₄ and CO₂) into syngas simultaneously; in addition, the DRM process is cheaper because it eliminates the complex gas separation of end products. Moreover, the ratio of H₂ to CO is close to 1:1, which is not only suitable for Fischer–Tropsch synthesis, but can also be used to produce high-value-added products such as methanol, acetic acid, dimethyl ether and oxyalcohol or long-chain hydrocarbons. Furthermore, syngas produced by DRM can be utilized for the storage of solar energy or nuclear energy, outperforming the other two reactions in terms of environmental and economic value [1–6].

The DRM reaction is a complicated process including several reactions:

$$CH_4 + CO_2 \rightarrow 2CO + 2H_2 \quad \Delta H^\theta_{298K} = 247.3 \ kJ/mol \tag{1}$$

$$CH_4 \rightarrow C + 2H_2 \quad \Delta H^\theta_{298K} = 75.0 \ kJ/mol \tag{2}$$

$$2CO \rightarrow C + CO_2 \quad \Delta H^\theta_{298K} = -171.0 \ kJ/mol \tag{3}$$

$$CO_2 + H_2 \rightarrow CO + H_2O \quad \Delta H^{\theta}_{298K} = 410 \text{ kJ/mol} \tag{4}$$

The main reaction (Equation (1)) is a very energy consuming (endothermic) reaction, requiring a high temperature to activate the reactants. Catalysts can be added to reduce the activation energy (Ea) required for the reaction, saving the energy input and accelerating the reaction without being consumed [2,4,7,8]. Coke formation mainly originates from methane decomposition (reaction 2) and CO disproportionation (reaction 3). At a relatively high temperature, reaction 3 is probably inhibited thermodynamically; however, reaction 2 is promoted, thus leading to more carbon deposition. Meanwhile, reversed water–gas shift (RWGS) (reaction 4) affects the $H_2$ selectivity. On the other hand, considering the strong C–H chemical bond and highest oxidation state of C in $CO_2$, the activation of $CH_4$ and $CO_2$ can be another challenge [9,10].

Noble metals (Rh, Ru, Pd, Pt, etc.) have features such as excellent performance and strong carbon deposition resistance in the DRM reaction [11,12]. For example, Ru-, Rh- and Pt-based catalysts possess a superior coke removal ability due to the high dispersion and small particle size [12]. However, limited by the high cost of precious metals, it is difficult to apply them on a large scale. In comparison, transition metals (Ni, Fe, Co, etc.) have drawn much interest because of their good catalytic performance and low cost. For example, Co has high affinity for the oxidative removal of carbon species [13]. In comparison, Ni is more reactive for $CH_4$ cracking and $CO_2$ activation [13]. Therefore, Ni-based catalysts have become promising for DRM reactions due to their rich reserves, low prices and high catalytic activity [14]. However, under high temperature and atmospheric pressure, carbon deposition and metal sintering lead to catalytic deactivation [14,15].

Strategies to improve the durability of Ni catalysts include enhancing the metal–support interaction (MSI) and generating reactive oxygen species. The former helps to inhibit metal sintering and greatly affects the catalytic performance of catalysts. The latter prevents or gasifies the carbon deposits. In addition, carbon deposition is closely related to the particle size of Ni. Due to the confinement effect of the support, the Ni particle size is reduced and the driving force of carbon diffusion in Ni crystals becomes weaker, significantly inhibiting the carbon deposition. Therefore, it is necessary to find a suitable support for Ni-based catalysts to enhance the metal dispersion, strengthen the MSI and introduce abundant oxygen defects [6]. Previous studies have shown that the coke formation can be reduced by adding metal oxides such as $Al_2O_3$, $SiO_2$, $TiO_2$ and $ZrO_2$ to enhance the MSI. Among them, alumina featured with its high specific surface area and appropriate pore size, exhibiting a stronger interaction with NiO and better catalytic performance compared with $SiO_2$, $TiO_2$ and $ZrO_2$ [10]. Zhang et al. [10] prepared Ni-based catalysts with $SiO_2$ and $Al_2O_3$ as the supports. The results showed that less carbon deposition was formed on the $Ni/Al_2O_3$ catalyst due to the stronger MSI between Ni and $Al_2O_3$ and the formation of smaller Ni particles. This was related to the formation of a $NiAl_2O_4$ spinel phase between the Ni precursor and $Al_2O_3$ during high-temperature calcination [16,17]. Moreover, oxygen vacancies can be generated in the reduced spinel structure, which promotes the adsorption and activation of $CO_2$ molecules to produce O· radicals to gasify the carbon deposits [17] (Table 1).

Despite the strong MSI and high surface area, the intrinsic acidity of $Al_2O_3$ activates the $CH_4$ decomposition but adversely affects the $CO_2$ adsorption, leading to a fast carbon deposition and a slow O· generation. The difference in reactant molecule activation rates results in coke formation and the coverage of Ni active sites [18]. In detail, the carbon nanotube formed in alumina firmly covers the Ni particles, causing the separation of the Ni active site from carbon dioxide and methane, and the weak interaction between Ni metal and $Al_2O_3$ by lifting Ni particles in the matrix, which subsequently leads to the migration of Ni particles to the outer surface of $Al_2O_3$ and easy agglomeration in the DRM reaction process [19]. To alleviate the above issue, Fahad S. Al-Mubaddel et al. [7,20] doped $La_2O_3$ into an acidic alumina support and found that the basic site density included a super basic site (related to monodentate carbonate) and low acid site density, which benefited the control of $CH_4$ dissociation and $CO_2$ chemisorption/dissociation on the catalyst surface

to oxidize the deposited carbon, greatly inhibiting the pulverization and graphitization of the carbon layer responsible for catalyst deactivation. However, another explanation was that on the acidic catalyst, methane is still activated on Ni, whereas carbon dioxide is dissociated on the catalyst support by forming carbonate/carboxylic acid.

Recently, strategies have been conducted to alleviate the carbon deposition and metal sintering of $Ni/Al_2O_3$ catalysts, including mesoporous alumina, morphology control, doping the second metal to form alloys, adding metal oxides to the support, reducing surface acidity, introducing oxygen defects and promoting surface oxygen mobility. However, reviews on related topics are still rare. Therefore, this paper will cover the latest research progress on the morphology and porosity control of alumina supports, surface acidity and alkalinity adjustments, alloy/MSI and oxygen defect formation, followed by the prospect of $Ni/Al_2O_3$ catalysts in DRM reactions in the future. It is believed that these strategies, proven effective in DRM reactions, will be promising in modifying the catalysts for other reactions such as steam reforming and tar reforming reactions, thus paving the way for a wide range of catalytic production techniques of syngas and clean energy.

## 2. Structure and Morphology Control

High-temperature thermal treatment in the DRM process will lead to the cracking of $CH_4$ and coke formation on the catalyst, and the sintering of metal particles, which covers the active sites and reduces the surface area, thus lowering the activity and stability of the catalyst. Therefore, an ideal support should enhance the metal dispersion and prevent carbon deposition [21]. Despite the admirable catalytic performance of $Ni/Al_2O_3$ catalysts in methane activation and $CO_2$ conversion, it also bears disadvantages such as coke deposition, sintering, phase transformation and reduction [21,22]. In 1992, a mesoporous material which can effectively control and reduce the metal deactivation rate was proposed for the first time, which is widely used in the field of catalysis [1]. Among them, mesoporous alumina, with a large surface area and ordered pore structures, anchors the metals within the structure and promotes metal dispersion, exhibiting better physico-chemical properties than non-porous alumina in terms of the surface area, pore size and thermal stability [23,24]. Due to the low cost and excellent properties, mesoporous alumina has attracted the attention of researchers and is widely used as a catalyst support [25]. For example, Farshad Gholizadeh et al. [1] prepared cubic-ordered mesoporous alumina (COMA) as the support for Ni by adjusting the pH, which successfully dispersed Ni evenly into alumina support. The ordered mesoporous arrangement in COMA improved the confinement effect on the $Ni^0$ active phase, i.e., it limited the well-dispersed Ni nanoparticles within the ordered mesoporous channels and imposed steric hindrance on $Ni^0$ species [1]. On the other hand, the ordered mesoporous structure provided a large surface area for the dispersion of Ni nanoparticles, which enhanced the dispersion of $Ni^0$ nanoparticles on the mesoporous support, thus strengthening the carbon deposition and sintering resistance of the catalyst [1,26]. The resulting conversion rates of methane and carbon dioxide were 93–99% and 91–97%, respectively, and the carbon deposition was only 5% in the 210 h stability test. In addition to the cubic mesoporous structure, a 2D hexagonal and mesoporous alumina exhibited similar effects on Ni particles. Karam Jabbour et al. [26] synthesized ordered and mesoporous $Ni/Al_2O_3$ through the "one-pot method". In addition to the steric hindrance, Ni nanoparticles were highly dispersed in the $Al_2O_3$ support. After 13 h of testing, most nanoparticles were still anchored in the pores of the support, greatly reducing the possibility of sintering and producing almost no crystalline carbon.

The mesoporous structure of alumina not only promotes the DRM reaction activity by improving the dispersion of Ni nanoparticles, but also enhances the carbon resistance of the catalyst by strengthening the interaction between the metal and support. The mesoporous alumina-supported Ni catalyst prepared by Bian et al. [27,28] via a surface impregnation method benefited the formation of well-dispersed Ni nanoparticles (Table 1). Additionally, all Ni existed in the form of $NiAl_2O_4$ after calcination at 700 °C. Due to the strong MSI in $NiAl_2O_4$ spinel structure, Ni crystal size was only about 5 nm after reduction. Small

particle size was proven effective in inhibiting the carbon deposition, which was conducive to the DRM reaction. Meanwhile, the collapse of pores in mesoporous alumina generated larger pores without adverse effects on the pore volume, which significantly enhanced the coking resistance and stability of the $Ni/Al_2O_3$ catalyst. At 700 °C, the conversion of methane and carbon dioxide reached 77.6% and 85.4%, respectively, and the carbon deposition was as low as 3.8%. Compared with the mesoporous structure, hierarchical porous structures possess a multi-functional catalytic effect. Ma et al. [19] prepared an alumina support with both mesoporous channels and a macroporous structure. Benefiting from the bimodal pore distribution, large surface area and Ni dispersion were achieved; on the other hand, abundant channels were provided for the diffusion of reactants and product molecules. Due to the hierarchical structure, the prepared $Ni/Al_2O_3$ catalyst exhibited both a high conversion and strong resistance to carbon formation [19,29,30].

Mesoporous structure plays an important role in stabilizing the metal particles in $Ni/Al_2O_3$ catalysts by improving the MSI and confinement effect. Additionally, a large surface area and pore size alleviate coking and sintering during the DRM reaction. Similarly, the morphology of the alumina support can also reduce the size and increase the dispersion of Ni, thus improving the catalytic performance. For example, the alumina in the form of nanofibers promoted the dispersion of supported Ni metal particles, reduced their size, enlarged the Ni active surface area (96 $m^2/g$), significantly inhibited the carbon deposition and sintering of Ni particles, increased the number of active sites, and facilitated the access of reactants into active nickel sites, positively influencing the stability and activity of $Ni/Al_2O_3$ catalyst [31,32]. In order to obtain better catalyst surface properties, ultrafine Ni particles and higher dispersion, Liang et al. [33] successfully synthesized raspberry-structured $Al_2O_3$ nanoparticles by the gas-phase method, which were well-mixed with the Ni precursor at acidic pH to form a precursor solution. Under evaporation-induced self-assembly (EISA), the stable $Al_2O_3$ nanoparticles gradually aggregated via capillary force to form ordered nanoparticle clusters (NPCs). The soluble Ni precursor was dried and deposited on the surface of $Al_2O_3$ NPCs. During thermal decomposition, the dried aerosol was transformed into nanostructures and NiO was formed in a single nanoparticle within the $Al_2O_3$ NPCs, followed by the transformation into Ni metal phase under $H_2$ thermal reduction. As shown in Figure 1a, ultrafine Ni crystals were homogeneously dispersed in $Al_2O_3$ NPCs. The Ni grain size was less than 7 nm and the metal surface area reached 129.9 $m^2/g$, much better than the controlled $Ni/Al_2O_3$ catalyst. Due to the uniformly disperse nanoscale Ni catalyst and strong MSI with the raspberry-shaped $Al_2O_3$, the sintering of Ni grains was prevented and mossy carbon deposits were easily removed from the Ni surface. The resulting conversion of $CH_4$ and $CO_2$ were both enhanced by 20–40% at various reaction temperatures ($T_{sur}$), as shown in Figure 1b,c.

In addition to nanofibers and clusters, two-dimensional alumina nanosheets promote the dispersion of Ni nanoparticles and a reduction in nickel size because of their high BET specific surface area, leading to stronger anti-sintering and anti-coking properties, and a significantly improved stability. The initial methane conversion was 56.0%, which remained stable after 300 min of reaction. Similar trends were observed in $CO_2$ conversion and the $H_2/CO$ ratio. The Ni nanoparticles were embedded into the alumina nanosheet, which triggered a strong MSI and enhanced the sintering resistance of the catalyst [34].

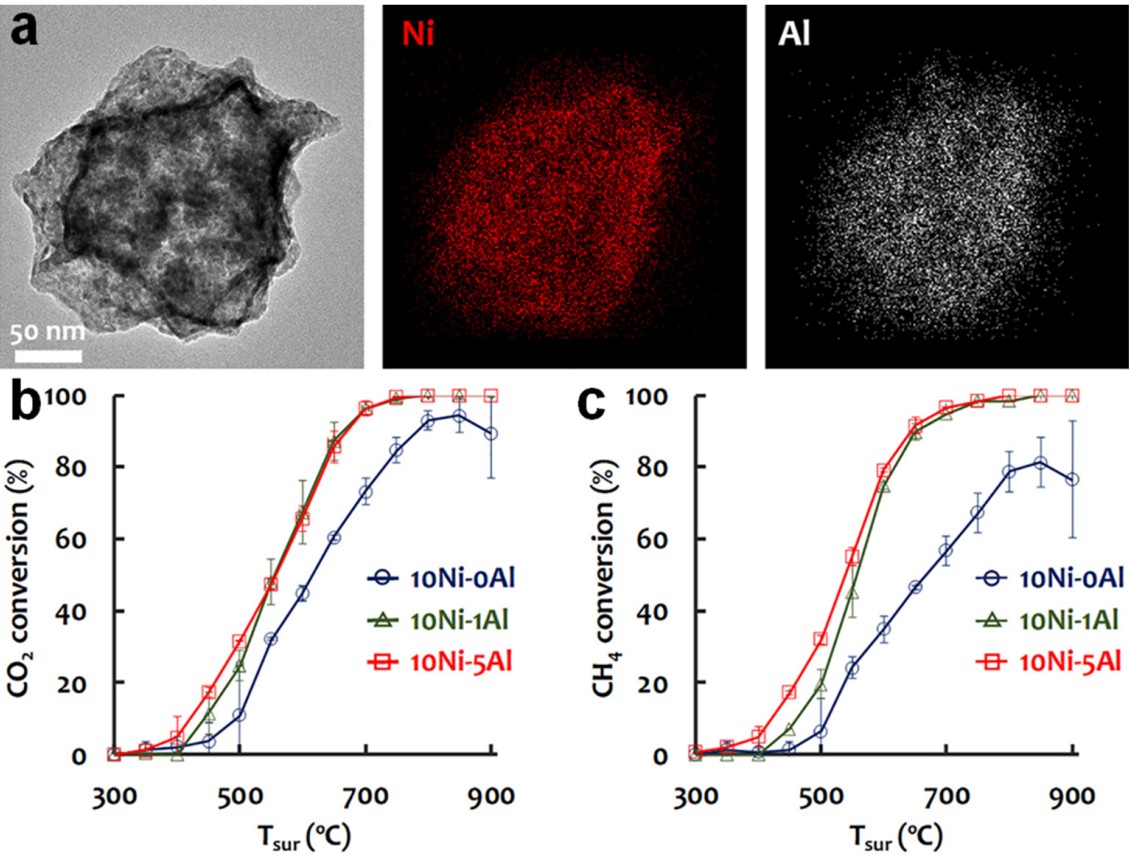

**Figure 1.** (**a**) HR-TEM image analysis of 10Ni-5Al with EDS elemental mapping. The scale bars are 50 nm. Activity tests of the catalysts using Ni-Al$_2$O$_3$ (**b**) $X_{CO2}$ vs. $T_{sur}$; (**c**) $X_{CH4}$ vs. $T_{sur}$. Reproduced with permission from [33]. Copyright 2020, Elsevier.

## 3. Surface Acidity/Basicity

In the DRM process, acidity and alkalinity are two of the key factors which determine the carbon on the catalyst surface. When Ni is supported on acidic alumina, the dissociation of methane is promoted so as to generate carbon deposition [28]. Increasing the basicity of the support through alkaline additives will enhance CO$_2$ chemisorption on the catalyst. On the one hand, enhanced basicity promotes the removal of coke formed in the process of CO disproportionation and methane cracking by oxidizing the carbon with CO$_2$, improving the stability of the catalyst. On the other hand, more basic sites facilitate the activation of CO$_2$ molecules, which is conducive to the catalytic conversion of reactants to syngas [35]. According to the methane reforming mechanism proposed by Tsipouriari et al. [36], the adsorbed CH$_x$ intermediate on the support reacted with the −OH group to form the formic acid intermediate CH$_x$O, which subsequently decomposed into H$_2$ and CO. The highly alkaline metal oxide support can provide more -OH groups, which not only enhance the syngas yield from CH$_x$O decomposition, but also promote the adsorption and dissociation of CO$_2$, so as to produce more oxygen radicals near the catalytic active metal surface and inhibit the formation of carbon [37,38]. In addition, the surface acidity and alkalinity affect the electronic environment of the Ni active site to realize carbon inhibition, lowering the rate and degree of methane dissociation and CO disproportionation [39].

Basic oxides can be added to Al$_2$O$_3$ to reduce the acidity of Ni/Al$_2$O$_3$ catalysts. For example, La$_2$O$_3$ can cover the acidic sites of alumina and increase the basicity of the catalyst, which benefits the chemical adsorption and dissociation of CO$_2$, and prevents carbon deposition in DRM reaction. The balance between acidic and alkaline sites on the catalyst surface can control CH$_4$ decomposition, coke oxidation by CO$_2$, and the inhibition of the graphitization of carbon deposits [18,40,41]. On the other hand, La$_2$O$_3$ can adsorb CO$_2$ on alumina supports to form intermediate carbonate (La$_2$O$_2$CO$_3$), which is an active species

and further reacts with carbon on the surface near nickel to produce CO and $La_2O_3$. This regeneration reaction of $La_2O_3$ reduces carbon deposition on the catalyst [42]. However, the generated $La_2O_2CO_3$ may lead to sintering and affect the stability of the catalyst.

Strong alkaline sites can enhance the $CO_2$ concentration on the catalyst surface, which not only increases the conversion rate of $CO_2$, but also accelerates the gasification of surface carbon and delays the generation of inactive carbon [43,44]. In addition to $La_2O_3$, MgO with a strong basicity can enhance the adsorption of $CO_2$ and its oxidation capability, thus reducing carbon deposition. Meanwhile, RWGS reactions could be inhibited, leading to a high $H_2$ selectivity. The Mg/Al ratio also determines the catalytic performance and deactivation behavior of $Ni/Al_2O_3$ in the DRM reaction. When the ratio increased from 0.1 to 0.24, lower acidity and more defects in the $MgAl_2O_4$ spinel phase were obtained, achieving a 1/3 carbon deposition rate. However, with a further increase in the ratio to 0.5, despite the stronger basicity, the loss of mesopores and lower surface area adversely affected the activity [45]. Therefore, if the support alkalinity is too strong, carbon deposition may not be inhibited, leading to an unstable conversion. Apart from adjusting the Mg/Al ratio, the addition of $Y_2O_3$ introduces more weak and medium alkaline sites into the $Ni/Al_2O_3$ catalyst (Table 1). Moreover, 1.5 wt.% $Y_2O_3$ optimized the surface basicity, exhibiting high and stable conversions of $CH_4$ (78.3%) and $CO_2$ (73.9%) [46,47].

In addition to metal oxides, P as a non-metal element can enhance the coke resistance and stability of $Ni/Al_2O_3$ in the DRM reaction by adjusting the acidity. The 2 wt.% P-modified $Ni/Al_2O_3$ catalyst displayed the lowest acidity and exhibited excellent anti-coking property in 100 h of reaction. When the basicity of the catalyst was increased, acidic carbon dioxide molecules were more likely to adsorb on the catalyst surface; therefore, the Boudouard reaction was promoted, which gasified the deposited carbon. Meanwhile, with the increase in P content, the generated $AlPO_4$ interfered with the interaction between the active Ni and $Al_2O_3$, so as to improve the reducibility of NiO to produce more active sites [48].

As well as the addition of basic oxides, the pretreatment atmosphere can affect the surface acidity and basicity. Compared with air calcination, Ar pretreatment could enhance the basic sites together with the addition of ethylene diamine tetra-acetic acid (EDTA) due to the promotional effect of carbon residues at the interface of $Ni^0$ and $La_2O_3$ [49]. In another case, $N_2$ calcination would increase the amount of Lewis acid sites on $Al_2O_3$, which strongly interacted with metallic Ni and reduced the Ni electron density, thus inhibiting metal sintering and carbon deposition [39,50–52].

Generally speaking, as an acidic support, $Al_2O_3$ induces methane cracking and slows down the activation of $CO_2$, thus resulting in carbon deposition. The acidity and basicity of the catalyst can be adjusted by adding alkaline metal oxides such as $La_2O_3$ and MgO or non-metal P and modifying the synthesis method, such as the pretreatment atmosphere, to promote the gasification of coke. However, excessive alkalinity will also lead to carbon deposition. Therefore, the basicity of the catalyst should be optimized to balance the activity and stability of $Ni/Al_2O_3$ in the DRM reaction.

## 4. Interfacial Engineering

By tuning the spatial and electronic environment via Ni alloy formation and adjusting the interaction between Ni and $Al_2O_3$ with the addition of another support, the metal agglomeration and coking process can be inhibited; additionally, the activation of $CH_4$ and $CO_2$ could be enhanced, thus leading to a good activity and stability in DRM reaction.

### 4.1. Alloy Formation

Ni-based catalysts will be deactivated due to the loss of active sites when sintering and carbon deposition occur in DRM reactions. By forming alloys with other metals such as noble metals (Pt, Rh, Ru) and transition metals (Fe, Co), the catalyst will possess a stable structure, enhanced reducibility and large specific surface area, resulting in a better performance than that of a single Ni catalyst. Additionally, alloy formation can lead to a

smaller metal particle size, which is conducive to the inhibition of carbon deposition and the gasification of carbon. For example, Ni and Co can form alloys to promote oxygen adsorption, prevent the adsorption of carbon, limit RWGS reactions and inhibit the sintering of Ni. As for Cu–Ni and Mg–Ni alloys, more active sites are generated on the surface along with the reduction in particle size and better dispersion [2,5,53]. The following sections will introduce the effect of Ni alloys with different metals on the catalytic performance of $Ni/Al_2O_3$ in DRM reactions.

### 4.1.1. Noble Metal–Ni Alloy

In DRM reactions, the selection of active metals plays an important role in inhibiting coking and enhancing the catalytic performance. Noble metals are widely used in reforming reactions and show better carbon deposition resistance than Ni in DRM reactions. Therefore, noble metals have been introduced into $Ni/Al_2O_3$ catalysts, forming a beneficial synergy. In addition, the bimetallic catalyst can promote the activation of C–H bonds in methane and maintain the constant conversion of methane at high temperatures.

It is well known that $Ni^{2+}$ ions are not active in methane reforming, and their coverage on the metal Ni surface reduces the active sites, resulting in a poor activation of methane [54]. Adding a small amount of Pt to Ni can inhibit the oxidation of Ni. This is because Pt cannot be oxidized below 300 °C, which stabilizes the Ni and prevents it from being oxidized. The addition of Pt also significantly lowers the reduction temperature because $H_2$ molecules overflow on the surface of Pt; after adsorption and dissociation, the generated H atoms react with NiO to form Ni metals. In addition, the interaction between Pt and Ni sites in the closed region and the dilution effect of Pt lead to a higher dispersion of Ni metal nanoparticles with a smaller size, resulting in a weak interaction with the tetrahedral structure of $CH_x$ species, which slows down the complete decomposition of $CH_4$ into coke [55]. At the same time, the Ni–Pt alloy promotes the formation of surface hydroxyl groups and improves the rate of carbon gasification [56,57]. Figure 2a shows that the optimal Pt loading was 0.4 monolayer, which exhibited the highest methane conversion due to the small alloy size. Meanwhile, it also presented an excellent sintering resistance (nearly no size change during the reaction). However, when the proportions of Pt continued to increase, larger particles were formed, resulting in higher activation energy (96.4 vs. 86.2 kJ/mol) and a poor activity (Figure 2a). On the other hand, when the calcination temperature was above 600 °C, separation of the Ni–Pt alloy occurred to produce large Pt ensembles (Figure 2b), which excessively promoted the decomposition of methane into surface carbon and intensified the deactivation rate of the catalyst [53].

In addition to the Ni–Pt bimetallic catalyst, the alloy formed by Rh and Ni presents high stability and strong anti-coking ability, which exerts a positive effect in DRM reactions [58–60]. Compared with $Ni/Al_2O_3$ catalysts, the interaction between Ni and Rh improves the conversions and $H_2$ yield with less coking and improved stability. This enhanced performance is due to the reduction in NiO species by high concentrations of spilled $H_2$ and the inhibition of CO disproportionation and $CO_2$ hydrogenation reaction [61]. Damyanova et al. [62] showed that the strong interaction between Rh and Ni not only improved the reducibility of NiO, but also increased the surface distribution of Ni, producing a small and uniform Ni particle (5 nm in diameter) evenly distributed on the $Al_2O_3$ support. In addition to the reducibility and particle size, the surface charge of the nanoparticles may also determine the activity. Due to the low surface energy of Rh relative to Ni, Rh atoms were preferably on the top of Ni layer. According to the Pauling scale, it could be inferred that negatively charged Rh and positively charged Ni coexisted in the Ni–Rh alloy due to the different electronegativities. During the reaction, $Ni^0/Ni^{n+}$ and/or $Rh^0/Rh^{n+}$ pairs may act as catalytic active sites, which probably enhanced the conversion of methane and the $H_2$ yield of $Rh–Ni/Al_2O_3$. Due to the above merits, the conversion of methane and carbon dioxide at 650 °C reached about 80% and 90%, respectively, with a high $H_2$ selectivity ($H_2/CO$ ratio = 1.79). Over a 3 h reaction test, no conversion drop

was shown for Rh–Ni/Al$_2$O$_3$, but a decrease in activity of over 10% was presented for Ni/Al$_2$O$_3$ [62].

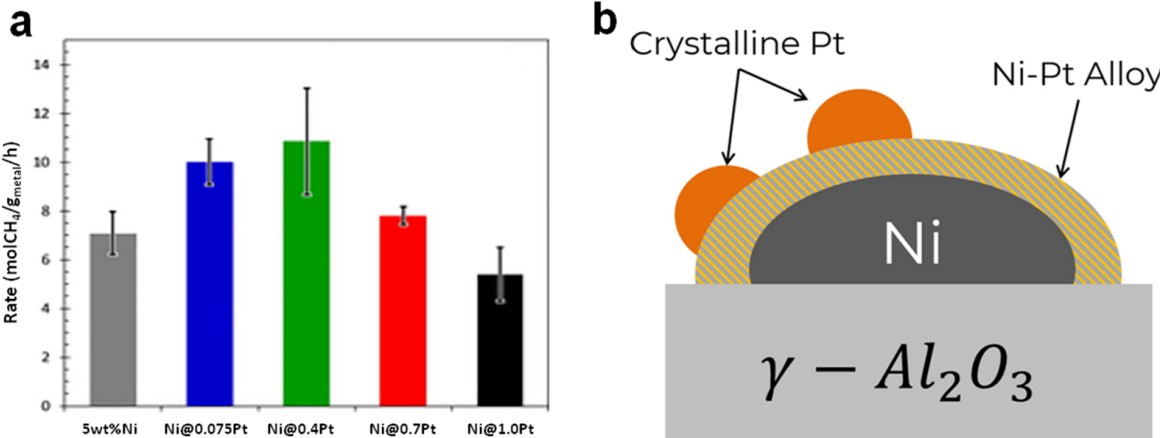

**Figure 2.** (**a**) Steady state CH$_4$ conversion rates at 550 °C. (**b**) Proposed structure for electroless deposition-derived Ni–Pt samples. Reproduced with permission from [53]. Copyright 2020, Elsevier.

In summary, for Ni/Al$_2$O$_3$ catalysts, due to the presence of oxygen, the surfaces of Ni particles are easily oxidized and covered with a layer of Ni$^{2+}$ ions, reducing the methane reforming activity. In contrast, the addition of noble metals into Ni/Al$_2$O$_3$ catalysts exhibits better catalytic performance due to the reducibility and intrinsic high activity of noble metals. In addition, noble metals and Ni exist in alloy states with a strong interaction, which effectively inhibits the agglomeration of nickel. Moreover, the generation of surface hydroxyl groups is beneficial for the removal of carbon deposition, so as to maintain the high activity and stability in DRM reactions.

### 4.1.2. Transition Metal–Ni Alloy

Although noble metals are good candidates for the formation of Ni alloys, the cost and scarcity limit their large-scale applications. In contrast, due to the satisfactory activity and low cost, transition metals have drawn much interest as dopants in Ni/Al$_2$O$_3$ catalysts in DRM reactions. Representative metals such as Cu, Fe and Co possess similar electronic structures and chemical properties to nickel, which facilitates their incorporation to form nickel-based alloys with synergistic effects, such as high dispersion, good activity and reduced coking tendency [63–65].

Among the transition metals, due to the similar lattice structure between CuO and NiO, the formation of a uniform NiO–CuO solid solution is promoted during calcination; after subsequent reduction in H$_2$, Cu can form alloys with Ni in Ni–Cu/Al$_2$O$_3$, which, in turn, stabilizes Ni and prevents the generation of large crystals. The Ni–Cu alloy shows excellent activities and effectively reduces the coking and metal sintering in DRM reactions. For example, Chatla et al. [66] doped Cu to Ni/Al$_2$O$_3$; the overlapping spatial distribution of Ni and Cu suggested the complete incorporation of copper into nickel lattices to generate Ni–Cu alloy particles, which directly tuned the electronic properties of the active metal sites. Additionally, the Ni–Cu alloy facilitated the H$_2$ spillover and effectively inhibited the formation of a NiAl$_2$O$_4$ spinel (lower temperature shift for the reduction temperature), improving the reducibility of metal and increasing the number of active sites (Figure 3a) [67]. Moreover, the average particle size of Ni–Cu alloy (4 nm) was smaller than that of single Ni catalyst (9 nm), indicating a higher dispersion of metal phase with the addition of Cu. Apart from the enhanced dispersion and reducibility, the higher activation barrier of dehydrogenation of CH$_4$ on medium content Cu catalysts suggested a less possibility of carbon formation. In addition, the carbon adsorption energy of single Ni catalyst was higher than that of Ni–Cu alloy, which indicated a lower surface

coking tendency of Ni–Cu/$Al_2O_3$. Moreover, with the Cu doping, the carbon elimination barrier was reduced, thus enhancing the ability of carbon gasification to carbon monoxide. Due to the promotional effects of medium Cu content in Ni–Cu/$Al_2O_3$, both the initial conversion of $CH_4$ and stability were enhanced (Figure 3b). Additionally, coke formation was effectively alleviated and most carbon formed was amorphous instead of graphitic, which was more easily gasified (Figure 3c). However, when the content of Cu was too high, the activation barrier of methane dehydrogenation would be greatly increased, hindering the process of the DRM reaction. Additionally, the Ni active site could be covered and was more prone to sintering [67]. Therefore, Cu content is crucial because it is related to the surface arrangement of Ni and Cu atoms. On one hand, the Cu content must be high enough to allow alloying with Ni; on the other hand, excessive Cu may cause segregation on the catalyst surface and the formation of Cu clusters, which will lead to the interruption of DRM reactions [68].

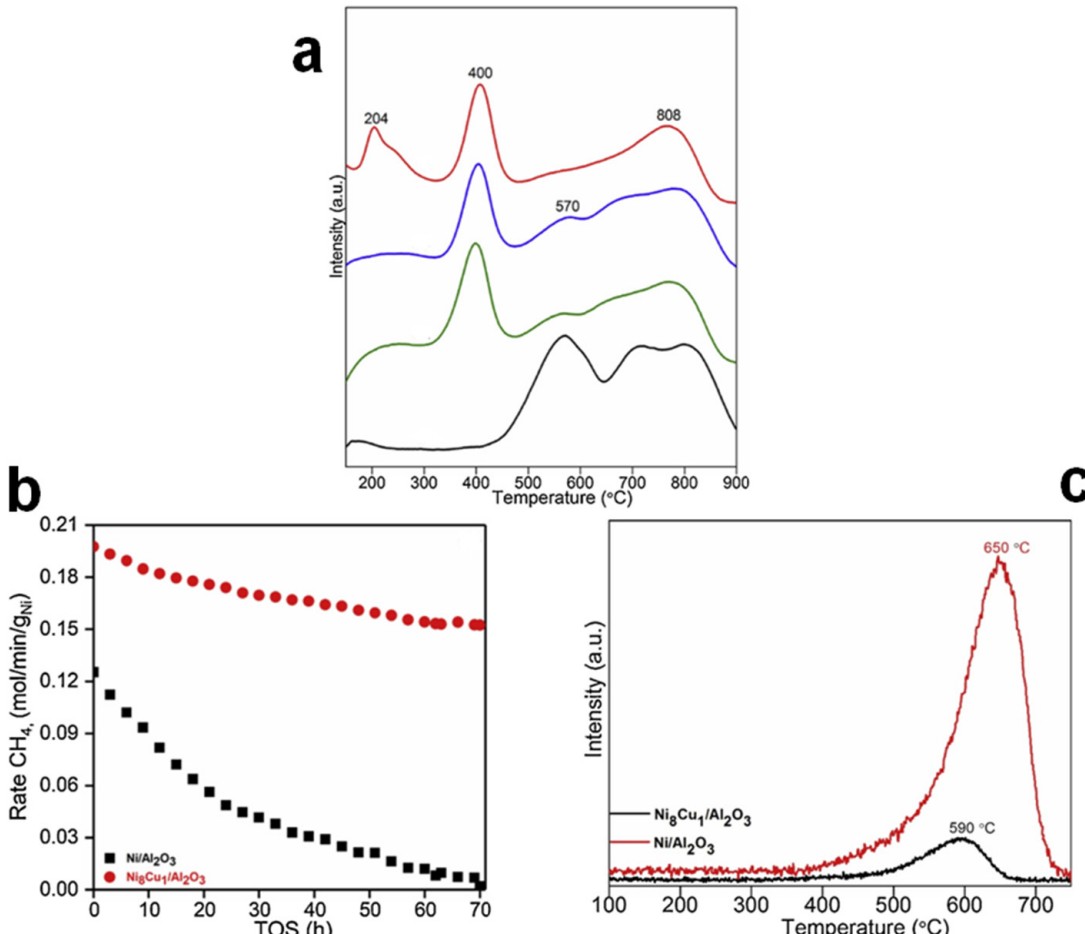

**Figure 3.** (**a**) $H_2$-TPR analysis of fresh calcined Ni/$\gamma$-$Al_2O_3$ (black), Ni10Cu1/$\gamma$-$Al_2O_3$ (green), Ni8Cu1/$\gamma$-$Al_2O_3$ (blue) and Ni3Cu1/$\gamma$-$Al_2O_3$ (red) samples. (**b**) $CH_4$ conversions rates vs. TOS for the $CO_2$ reforming of methane at 650 °C. (**c**) $O_2$-TPO profile of spent catalysts after DRM performance at 650 °C for 70 h TOS. Reproduced with permission from [66]. Copyright 2020, Elsevier.

Although Ni–Cu shows excellent ability in reducing carbon deposition, Ni–Co catalysts present higher metal dispersion than Ni–Cu. By comparing the promotion effects of Cu and Co on Ni/$Al_2O_3$ catalyst, the Ni–Cu active sites are unevenly dispersed due to the sintering of copper particles in the calcination stage. On the contrary, uniform morphology, high surface area, large pore size and porosity, and well-dispersed particles can be achieved in Ni–Co/$Al_2O_3$ nano catalyst [22,65,69]. These metal particles are con-

fined in the porous structure of alumina matrix, which helps to resist particle aggregation, in turn improving the resistance to carbon deposition [22]. Liu et al. [63] synthesized a Ni–Co/$\gamma$-Al$_2$O$_3$ bimetallic catalyst; the addition of Co increased the metal dispersion by 12 times. In addition, the higher reduction temperature in TPR profiles of Ni–Co/Al$_2$O$_3$ (Figure 4a) suggested a stronger MSI, which anchored the active metal particles in the cavities, enhancing the sintering-resistance and forming a homogeneous size distribution, thus improving the activity and stability in DRM reactions [13,22,65]. On the other hand, the existence of Co led to the formation of new active sites and the enhancement of the alkalinity of bimetallic catalysts, as shown in the higher peak intensity in Figure 4b, which benefited the adsorption CO$_2$ molecules and accelerated the oxidation of surface carbon species. Benefiting from the above properties, as shown in Figure 4c,d, the catalytic activities of the Ni–Co alloy catalyst were higher than those of the single metal counterparts, and the conversion rates did not drop obviously after testing at 700 °C for 3 h, which could be attributed to the high metal dispersion, a large number of active sites, abundant basic sites and strong MSI. In order to further clarify the synergistic effect of Ni–Co alloy, Aghaali et al. [22] prepared a well-dispersed Ni–Co alloy on an Al$_2$O$_3$ support (Table 1). Co possessed a high affinity for carbon species and promoted the removal of carbon by oxidation; Ni was more active for methane cracking. The strong Ni–Co bimetallic interaction not only promoted the Boudouard reaction and the oxidation of adsorbed CH$_x$ species, but also enhanced the H$_2$ yield. The resulting carbon deposition was only 1.3 wt.%, which could be easily removed by oxidation.

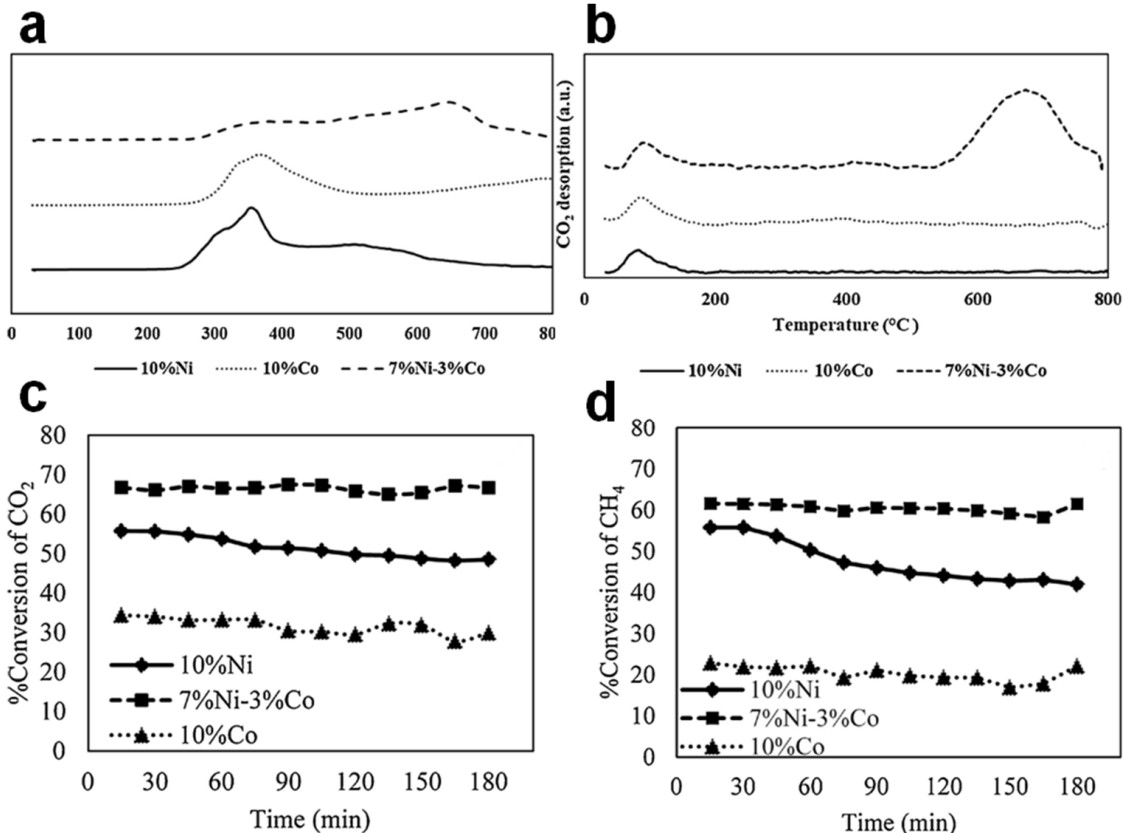

**Figure 4.** (**a**) The TPR profiles of the nickel–cobalt monometallic and bi-metallic catalysts over $\gamma$-Al$_2$O$_3$-HY zeolite. (**b**) CO$_2$-TPD profiles of the nickel–cobalt monometallic and bi-metallic catalysts over $\gamma$-Al$_2$O$_3$-HY zeolite. Catalytic performance of (**c**) CO$_2$ conversions and (**d**) CH$_4$ conversions. Reproduced with permission from [63]. Copyright 2020, Elsevier.

In addition to Cu and Co, Fe is active and is widely used in alloy catalysts because of its good redox ability [70]. Ray et al. [71] found that Fe and Ni formed FeNi$_3$ alloys whose H$_2$ chemisorption concentration was greater than that of Ni–Co/Al$_2$O$_3$. Despite the

higher activities of Ni–Co/Al$_2$O$_3$, Ni–Fe/Al$_2$O$_3$ exhibited a lower deactivation rate and better stability. Meanwhile, the carbon deposition rate was slower in Ni–Fe/Al$_2$O$_3$ than that in Ni–Co/Al$_2$O$_3$ [71]. Therefore, Ni–Fe/Al$_2$O$_3$ could be a promising DRM catalyst. Fe and Ni have similar electronic structure and chemical properties; therefore, it is easy for Fe and Ni to react with each other to form a bimetallic Ni–Fe alloy after reduction. The synergistic effect can reduce the sintering and carbon deposition in DRM reactions. Li et al. [72] synthesized a Ni–Fe alloy supported on ordered mesoporous alumina through the "one pot" evaporation-induced self-assembly (EISA) method (Table 1). The metal sites were confined in the alumina pores, which minimized the metal migration and sintering. Due to the strong redox properties of Fe, the oxygen storage capacity of the catalyst was enhanced so as to remove carbon deposition. To further elucidate the anti-coking mechanism, Kim et al. [70] synthesized a Ni–Fe/Mg$_x$Al$_y$O$_z$ catalyst, in which iron migrated onto the surface of the support by FeO/Fe redox cycle. During the reaction, Fe was partially oxidized to FeO, resulting in partial de-alloying and the formation of a Ni-rich Ni–Fe alloy. The generated FeO was located on the surface of the catalyst and covered a small part of Ni-rich particles. The intimate contact of FeO and nearby Ni–Fe facilitated the coke oxidation and reduction of FeO into the Ni–Fe alloy through a redox mechanism, which significantly inhibited the carbon deposition and improved the stability of the Ni–Fe/Al$_2$O$_3$ catalyst.

### 4.1.3. Ni–Sn Alloy

In addition to noble metals and transition metals, other main-group metals such as Sn exhibit good catalytic activities in DRM. The addition of a small amount of Sn to Ni/Al$_2$O$_3$ can obtain a bimetallic catalyst with a higher activity, reduced carbon deposition and improved stability compared with a single Ni catalyst. Da Silva et al. [73] found that due to the interaction between Ni and Sn, this bimetallic catalyst was excellent in the activity, stability and selectivity of syngas production (Table 1). Regarding carbon deposition, the Ni–Sn alloy could promote the oxidation of carbon on the catalyst surface. With a similar electronic structure to carbon, interactions between the Sn 5p orbital and Ni 3d electrons were strengthened. The change in electronic structure affected the reactant adsorption and hindered carbon nucleation on the active Ni sites, thus reducing the possibility of nickel carbide formation as the coke precursor and avoiding the diffusion of carbon to form large coke aggregates. After the DRM reaction, the surface of the Ni/Al$_2$O$_3$ catalyst was covered by graphitized carbon, whereas for Ni–Sn/Al$_2$O$_3$, only part of the catalyst surface was covered by filamentous carbon, which was softer and easier to remove, doing little harm to the deactivation of the catalyst. Therefore, the bimetallic catalyst still exposed most of the active sites after continuous operation, enhancing the stability for DRM reactions [74,75]. In addition to the anti-coking performance, the addition of Sn improved the anti-sintering ability, increased the metal active sites, and enhanced the yield of syngas, especially H$_2$ yield. Due to the gasification of residual carbon during the reaction, the volume originally occupied by carbon became pores, where uniform Ni–Sn alloy particles were highly dispersed, which could reduce the metal sintering. In addition, the affinity between H and Ni became weak due to the existence of Sn, suggesting an easier desorption of H$_2$ from the surface of the alloy, which inhibited the side reactions such as RWGS, resulting in fewer by-products and higher H$_2$ yield. On the other hand, Sn promoted the oxidation of reaction intermediates such as CHO, benefiting the formation of the final product—syngas [74]. However, the activity of the catalyst was affected by the content of Sn. A lower loading of Sn enabled an improved activity and reduced the carbon deposition. If excessive Sn was doped, the increased electron density originating from the hybridization of the Ni 3d orbital and Sn 5p orbital greatly hindered the activation of CH$_4$, thus leading to a low catalytic activity in DRM reactions [73].

### 4.2. Metal–Support Interaction (MSI)

A metal–support interaction (MSI) helps to anchor nickel particles on the support to inhibit their sintering, modifying the metal size, dispersion, surface area and reducibility of the nickel catalyst, thus enhancing the activity and stability of the DRM reaction [76,77]. For example, the unique stability of the $Ni/Ce–Al_2O_3$ catalyst was attributed to the good morphology and sufficient MSI, which stabilized the nickel species on the $Al_2O_3$ support and inhibited coking by promoting the Boudouard reaction [78]. In addition, the introduction of La prevented the sintering of Ni under harsh reaction conditions by enhancing the interaction between Ni metal and the $Al_2O_3$ support, leading to improved activity and stability [18]. However, a high temperature is needed to reduce the metals strongly interacting with the support, which may cause metal sintering under an intensified thermal treatment and the loss of active surface area for the adsorption and activation of reactant molecules in the DRM reaction. Thus, an optimized MSI will be appropriate considering the reducibility and stability of metal sites on the support.

Ni can easily react with $Al_2O_3$ to form a $NiAl_2O_4$ spinel at a temperature above 600 °C. The formation of this aluminate enhances the MSI and shows good anti-sintering and anti-coking properties in the DRM process. In terms of size control, the Ni size reduced from $NiAl_2O_4$ was 11 nm smaller than that derived from naked $Al_2O_3$, suggesting that the presence of $NiAl_2O_4$ stabilized the Ni particles at a high temperature. In addition, the incomplete reduction of $NiAl_2O_4$ produced a $Ni_xAl_yO_z$ support, which was considered a defective $NiO–Al_2O_3$ solid solution species with multiple $Ni^{2+}$ defects, also stabilizing the size of Ni. In terms of carbon deposition, the coke formed on $Ni/Al_2O_3$ produced by $NiAl_2O_4$ was only 8 wt.%, much lower than that on the Ni reduced from NiO in $Ni/Al_2O_3$ (37 wt.%) after 100 h of DRM. This difference in coke resistance could be explained by the fact that the easily reducible NiO species produced Ni sites with weak MSI; on the contrary, the partial reduction and extraction of $Ni^{2+}$ ions in $NiAl_2O_4$ were expected to possess sufficient interaction with $Al_2O_3$ to inhibit carbon deposition. In addition, in a reductive atmosphere, $NiAl_2O_4$ formed a defective $Ni_xAl_yO_z$ phase, which was likely to exhibit high surface oxygen mobility. Therefore, the high stability of $NiAl_2O_4$ could also be related to the high carbon gasification rate [76]. However, the $NiAl_2O_4$ spinel needs a high temperature to be reduced to NiO and Ni due to strong MSIs. Thus, the amount of active Ni may be limited, and high-temperature metal sintering can occur. Therefore, the formation of a $NiAl_2O_4$ spinel may lead to the deactivation of the $Ni/Al_2O_3$ catalyst in the methane reforming reaction [79,80]. In order to reduce the formation of $NiAl_2O_4$ and alleviate the excessive MSI, other metal oxides can be added to reduce the interaction between Ni and $Al_2O_3$. For example, when calcium was added to $Al_2O_3$, calcium competed with nickel to interact with alumina and tended to form calcium aluminate, reducing the possibility of the subsequent formation of nickel aluminate and producing more reducible Ni species, as shown in the lower temperature shift of the reduction peak in TPR profiles (Figure 5a). Meanwhile, Ca could increase $CO_2$ adsorption and activation, increase methane conversion and reduce carbon deposition. However, a further increase in calcium content would decrease the specific surface area and increase the electron density on the surface of nickel, which reduces the adsorption of methane molecules and decreases the conversion (Figure 5b) [81].

In addition to Ca, rare earth metals can also adjust the MSI. The presence of Ce helps to increase the dispersion of metal nickel with a strong MSI, inhibiting the growth of carbon whiskers and metal sintering, and generating a smaller Ni particle size [82]. Unlike ceria, which tends to exist in the form of crystals on the catalyst surface, $La_2O_3$ presents a monolayer dispersion, possessing a strong interaction between $La_2O_3$ and alumina. Additionally, the introduction of La prevents the sintering of Ni under harsh reaction conditions by enhancing the interaction between Ni metal and support. However, the monolayer $La_2O_3$ will cover the Ni active sites, resulting in reductions in the active specific surface area. Li et al. [18] doped La into $Ni/Al_2O_3$ via a novel synthesis method. By calcining $La_2O_3$ on $Al_2O_3$ in a $CO_2$ atmosphere, $La_2O_2CO_3$ was generated to replace $La_2O_3$,

so as to prevent Ni from entering the bulk phase of $Al_2O_3$ to form the spinel phase, improve the reduction degree of nickel and increase the specific surface area. Compared with blank $Ni/Al_2O_3$, La-modified $Ni/Al_2O_3$ showed higher activities in the DRM reaction (Table 1).

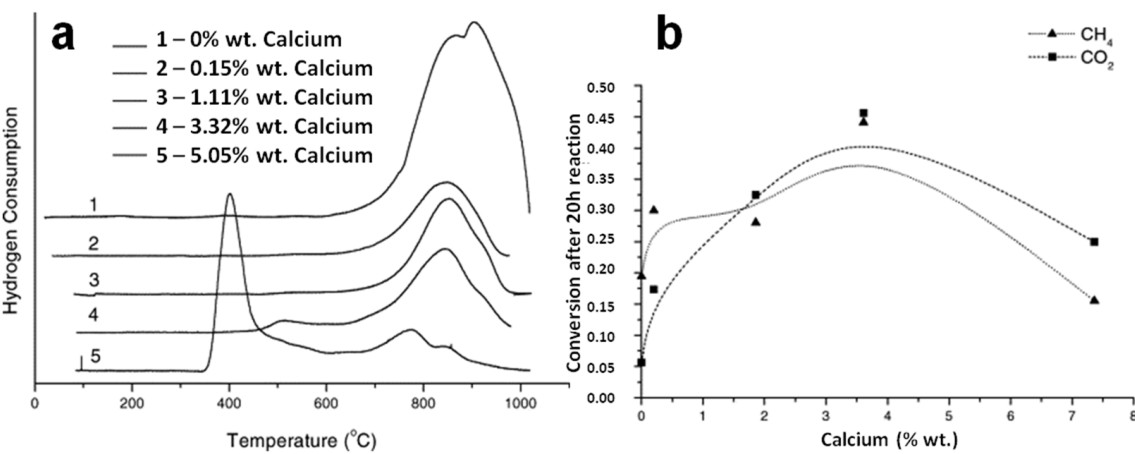

**Figure 5.** (**a**) Temperature-programmed reduction profiles of samples with addition order 1Ca2Ni. (**b**) Change in total methane and carbon dioxide conversion after 20 h reaction with amount of calcium of 1Ni2Ca. Reproduced with permission from [81]. Copyright 2003, Elsevier.

## 5. Oxygen Defects and Surface Oxygen Species

It is well known that lattice/surface oxygen and oxygen vacancy enhance the mobility and adsorption of oxygen and effectively remove carbon deposition in the DRM reaction [83]. Due to the high reducibility, facilitated oxygen mobility and the redox cycle of $Ce^{4+}/Ce^{3+}$, $CeO_2$ is reversibly converted to non-stoichiometric oxides, thus providing high oxygen storage capacity and oxygen mobility [84–87]. The redox pair is formed due to the electronic interaction between $CeO_2$ and Ni. Ce is rich in d-orbital electrons whereas Ni has unfilled d-orbitals; thus, the unfilled d-orbital of Ni atom can accept the d-electrons of Ce. Under a reducing atmosphere, $CeO_2$ can be converted to the mixture of $Ce_2O_3$ and $CeO_2$. The catalytic mechanism of this redox pair on the DRM reaction is as follows: the reactant $CO_2$ first adsorbs on the basic sites, followed by dissociation on $Ce_2O_3$ to produce CO and $CeO_2$ through the transfer of electrons. Then, $CeO_2$ reacts with the deposited carbon produced by $CH_4$ dehydrogenation, transforming back into $Ce_2O_3$ again. Meanwhile, due to the dissociation of $CO_2$, the adsorbed oxygen atoms and other oxygen-containing species are responsible for inhibiting the formation of carbon. Therefore, the presence of $CeO_2$ and $Ce_2O_3$ in the catalyst will improve the inhibition of surface carbon formation [86,88]. In another explanation, ceria can enhance the reversible oxygen adsorption/release capacity of $Ni/Al_2O_3$ catalysts. With the flow of surface oxygen species, $CO_2$ molecules are adsorbed to form bidentate carbonate, which react with surface carbon species, resulting in a higher $CO_2$ conversion and lower carbon formation [35,86].

Based on the above mechanism, Chen et al. [89] prepared a series of $Al_2O_3$-supported Ni catalysts modified with different amounts of $CeO_2$ by incipient wet impregnation (Table 1). As shown in Figure 6a, under operating conditions, Ce existed as a mixture of $Ce^{3+}$ in $CeAlO_3$ and $Ce^{4+}$ in $CeO_2$. Due to the lower oxidation state and abundant oxygen vacancies, $CeAlO_3$ adsorbed and dissociated $CO_2$ to generate CO and active oxygen species, which formed an oxidative environment around nickel sites, resulting in the gasification of carbon atoms at the nickel–support interface immediately before the nucleation and growth of the graphene layer, greatly improving the carbon resistance of the catalyst. Subsequently, the previously generated $CeO_2$ reacted with $CO_2$ to produce carbonate ions, which reacted with $CH_x$ derived from the $CH_4$ decomposition on the nickel surface to further improve the yield of syngas and carbon inhibition. As shown in Figure 6b, with the increase in Ce content, the carbon deposition decreased sharply, and the proportion of graphitized

carbon also reduced, suggesting an easier removal of coke. Quantitatively, when the Ce content was 15 wt.%, the carbon deposition was only 0.29 $g/g_{cat}$; moreover, the addition of Ce did not affect the conversion of $CH_4$ and $CO_2$, and maintained the dispersion and activity of Ni sites (Figure 6c) [89]. To further improve the oxygen mobility, $ZrO_2$ can be added to $Ni/Ce–Al_2O_3$. The formation of $CeO_2/ZrO_2$ solid solution generated more oxygen vacancies, increased the reducibility and enhanced the oxygen storage capacity, thus reducing the coke formation and improving the catalytic stability [90].

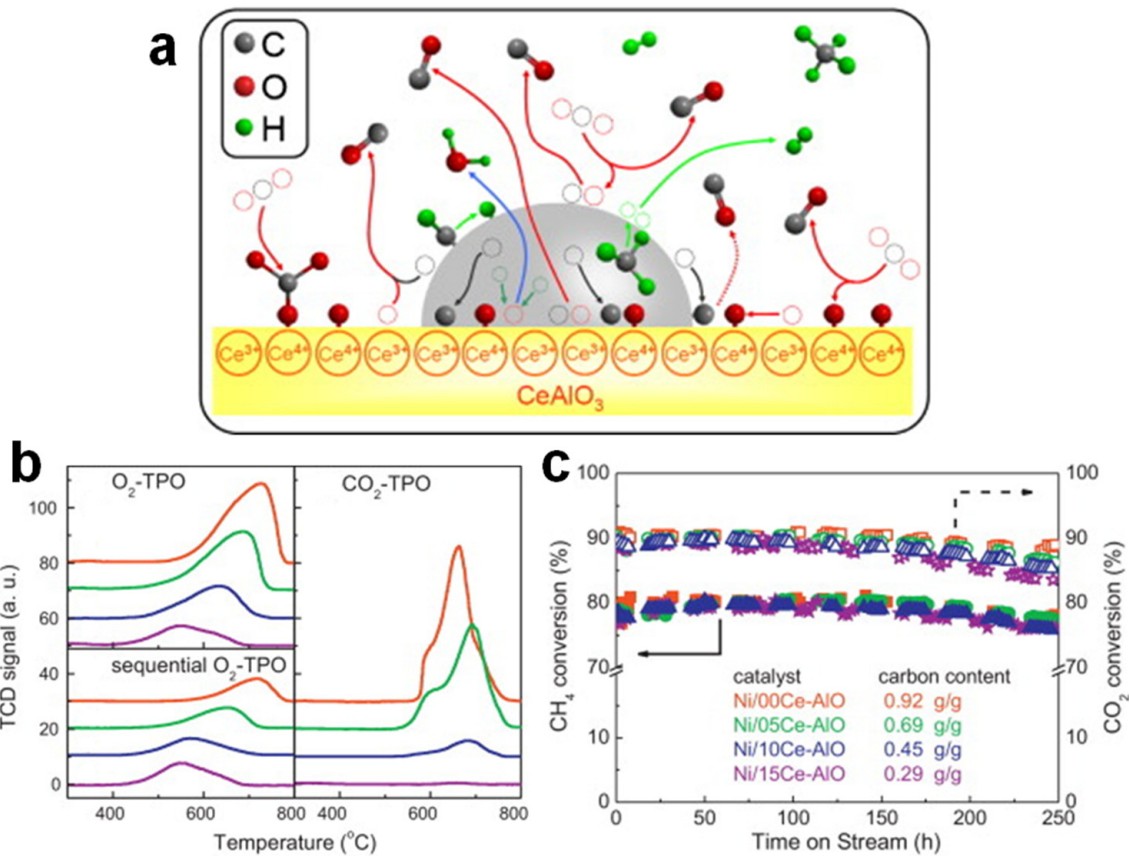

**Figure 6.** (**a**) Schematic representation of reaction and carbon deposition over Ni/Ce–AlO catalysts. (**b**) $O_2$-TPO, sequential $CO_2$-TPO and $O_2$-TPO profiles of the catalysts after 250 h DRM tests at 800 °C: Ni/00Ce–AlO (red), Ni/05Ce–AlO (green), Ni/10Ce–AlO (blue) and Ni/15Ce–AlO (purple). (**c**) Activity performance and carbon content of the catalysts in 250 h long-term tests. Reaction conditions: 800 °C, GHSV = 20,000 mL $min^{-1}$ $g^{-1}$, atmospheric pressure. Square: Ni/00Ce–AlO; circle: Ni/05Ce–AlO; triangle: Ni/10Ce–AlO; star: Ni/15Ce–AlO. Reproduced with permission from [89]. Copyright 2013, Elsevier.

Other rare earth metals also exhibit excellent thermal stability, redox potential and good oxygen storage capacity. For example, Sm doping in $Ni/Al_2O_3$ lowers the reduction temperature and enhances the reducibility of Ni nanoparticles from $Al_2O_3$. Similarly to Ce, the redox property of Sm promotes oxygen transfer and $CO_2$ adsorption. In addition, the intrinsic high oxygen vacancy and the extra oxygen vacancies generated by the exchange of $Ni^{2+}$ and $Sm^{3+}$ ions facilitate the adsorption of $CH_4$, increasing the conversions and inhibiting coke formation in the DRM reaction [91]. Another rare earth metal oxide $Y_2O_3$, featuring excellent oxygen mobility and redox properties, can minimize the accumulation of different carbonaceous species on the spent catalyst surface by improving the coke gasification rate, leading to improved activity and stability [92]. In addition to Ce and Y, the addition of $La_2O_3$ to $Ni/Al_2O_3$ can control the decomposition of $CH_4$, the oxidation of carbon deposits by $CO_2$ and the inhibition of the graphitization of carbon deposits [40]. Figueredo et al. [93] prepared a $Ni/La–Al_2O_3$ catalyst and found that the addition of La

could stabilize the $\gamma$-$Al_2O_3$ structure and form perovskite $LaAlO_3$, possessing abundant oxygen vacancies and high structural stability. The oxygen vacancy in $LaAlO_3$ perovskite reduced the MSI and increased the metal active sites. Meanwhile, the high oxygen mobility of perovskite promoted the activation of C–H bonds, resulting in the catalytic activity of Ni for methane conversion [75,93].

**Table 1.** Summary of the $Ni/Al_2O_3$ catalysts in DRM reaction.

| Catalyst | Preparation Method | Reaction Condition | | CH₄ Conversion | CO₂ Conversion | Carbon Formation | Reference |
|---|---|---|---|---|---|---|---|
| | | $CO_2/CH_4$ | Reaction Temperature | | | | |
| FeNiAl catalyst | one-step EISA method | 1 | 700 °C | 72.5% | 82.3% | 3.8% | [72] |
| $Ni_{0.05}Al_1O_{2-\delta}$ | citric acid sol–gel method | 1 | 650 °C | 68.7% | 80.4% | 0.59 $mg_C\ g_{cat}^{-1}\ h^{-1}$ | [17] |
| $Ni/MgAl_2O_4$ | wet impregnation | 1 | 750 °C | 96% | 98% | 8.95% | [83] |
| $Ni/M–Al_2O_3$ | incipient impregnation method | 1 | 700 °C | 77.6% | 85.4% | 3.5% | [26] |
| 15%NiO/COMA | sequential impregnation | 1 | 850 °C | 98% | 97% | 1.3 g/ml | [1] |
| $Ni/15\%CeO_2–Al_2O_3$ | incipient wetness impregnation | 1 | 800 °C | 80% | 85% | 0.29 $g\ g_{cat}^{-1}$ | [89] |
| $Ni/La_2O_3–Al_2O_3$ | stepwise incipient wetness impregnation. | 1 | 650 °C | 61% | 65% | Less than 4% | [18] |
| $Ni/Y–Al_2O_3$ | co-precipitation | 1 | 700 °C | 74.4% | 78.6% | Less than 12% | [47] |
| Ni supported on mesoporous alumina | incipient impregnation method | 1 | 700 °C | 77.6% | 85.4% | 3.8% | [27] |
| $Ni/Al_2O_3$ | solution combustion synthesis | 1 | 800 °C | 87% | 94% | 0.0378 $g_C\ g_{cat}^{-1}\ h^{-1}$ | [80] |
| $Ni–Co–Ru/MgO–Al_2O_3$ | neutral sol–gel | 1 | 800 °C | 93.2% | 92.5% | 8.1 $mg_C\ g_{cat}^{-1}\ h^{-1}$ | [65] |
| $Ni_{0.8}Co_{0.2}Al_2O_4$ | ultrasonic spray pyrolysis | 1 | 750 °C | 95% | 91% | 18.2% | [22] |
| 10Ni–2Sn/$Al_2O_3$ | modified Pechini method | 1 | 650 °C | 33% | 35% | 7.2% | [73] |

## 6. Conclusions and Prospects

In this review, several strategies of enhancing the coking and sintering resistance of $Ni/Al_2O_3$ in DRM reactions have been discussed in depth. Smart designs of hierarchical porous structures and 1D/2D nanomaterials can promote Ni dispersion and enlarge the surface area, thus preventing the metal agglomeration and increasing the number of active sites. Additionally, optimizing the surface acidity/basicity of alumina by adding alkali/alkaline metal oxides enables a better adsorption and activation of $CO_2$, promoting the syngas production and carbon removal. Moreover, the synergistic effects of Ni alloys and appropriate strength of the MSI hinder the migration of Ni and control the reducibility. Finally, the introduction of oxygen defects and generation of active surface oxygen species by adding rare earth metal oxides facilitate the oxygen movement and gasification of coke. Despite some progress made in related fields, there is still room for improvement, as discussed below.

Firstly, most of the modifications are limited to one or two functions. Certain improvements have to be at the expense of affecting other properties or performances. For example, the addition of Sn could slow down methane cracking and inhibit coke formation; however, coverage on the Ni active sites may reduce the conversions. In this situation, multi-functional catalysts are suggested to be developed to enhance the activity, stability and anti-deactivation properties simultaneously by taking advantages of the synergistic effects of each component in the catalyst. For instance, when mesoporous $La_2O_3$ is doped in $Ni/Al_2O_3$, the Ni dispersion, surface basicity, MSI and oxygen defects can be adjusted to a suitable level, achieving a robust and active catalyst.

Secondly, the specific role of the $NiAl_2O_4$ spinel phase is still in debate; either a promotional effect on the MSI or a negative impact on the metal active sites has been reported for DRM reactions. More systematic and mechanism studies are recommended to elucidate the exact formation process and the influences of the resultant spinel structure on the catalytic performance and deactivation behavior.

Thirdly, excessive basicity and MSI may not benefit the performance. Thus, investigations on the number of promoters (e.g., MgO) are recommended, as well as the thermal treatment parameters, in order to achieve a suitable degree of basicity and MSI for the optimized adsorption/activation of $CO_2$ molecules and generation of metal active sites for $CH_4$ dissociation.

Finally, understanding the deactivation mechanisms of $Ni/Al_2O_3$ lays the foundation for modifications of the development strategies. More efforts are expected to explore the coke generation and removal process and metal migration and agglomeration by taking advantage of advanced characterization techniques (e.g., in situ monitoring systems).

**Author Contributions:** Conceptualization, data curation, investigation, writing—original draft, X.G., Z.G., G.Z., and Z.W.; writing—review and editing; X.G. and Z.G.; project administration, supervision, validation J.A. and S.K.; funding acquisition, resources, S.K. All authors have read and agreed to the published version of the manuscript.

**Funding:** This research was funded by the Ministry of Education in Singapore (MOE) Tier 2 grant (WBS: R279-000-544-112), Singapore Agency for Science, Technology and Research (A*STAR) AME IRG grant (No. A1783c0016), Guangzhou Basic and Applied Basic Research Project in China (202102020134) and Youth Innovation Talents Project of Guangdong Universities (natural science) in China (2019KQNCX098).

**Data Availability Statement:** All data included in this study are available upon the permission from the publishers.

**Conflicts of Interest:** The authors declare no conflict of interest.

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
