# Peer review of "Anti-Coking and Anti-Sintering Ni/Al2O3 Catalysts in the Dry Reforming of Methane: Recent Progress and Prospects"

_catalysts, doi:10.3390/catal11081003_

Round 1

Reviewer 1 Report

Several strategies for hindering coke deposition and metal sintering in the DRM have been presented in this review. The manuscript is well-structured, and its content is clearly demonstrated with innovation. I believe that this study is of great relevance to the readers of Catalysts and I recommend its publication after some minor revisions are carried out.

  • This review is focused on Ni/Al2O3 catalysts. Why have the authors selected this specific catalyst for the DRM? Could the strategies proposed for avoiding metal sintering and coke deposition be similar to other catalysts?
  • The authors have provide a good overview of the most influential features (acidity, specific surface area, metal dispersion…) that affects the performance of Ni/Al2O3 catalyst on the DRM. However, the suitable catalyst design of Ni/Al2O3 catalyst is not specific of DRM, but also is common to other processes such as steam reforming reaction. The authors are suggested to include it in the text.
  • In Section 3, the influence of acidity/basicity on the coke deposition is established. However, in this section, the only way presented to modulate Ni/Al2O3 acidity is the incorporation of alkaline metal oxides. The authors are suggested to include how catalyst synthesis and Al2O3 pretreatment may affect catalyst acidity. Thus, it is well-established that the acidity of the Al2O3 support is related to the crystalline phase of the Al2O3 support. Thus, the acid sites of γ-Al2O3 promote carbon deposition on the catalyst surface, which is the main cause of catalyst deactivation in reforming reactions. [https://doi.org/10.1021/acs.energyfuels.1c01666].
  • In order to complete Table 1 with the main points addressed in the review, the authors are suggested to include more studies in which Ni/Al2O3, Ni-M/Al2O3 (being M noble metal, transition metals or Sn) are used in the DRM.
  • In Section 6: Conclusions and prospects, the contents about shortcomings and future development are not adequately covered in conclusion. The authors are suggested to expand this section in terms of the main challenges and more detailed directions for future research.

Author Response

Responses to Reviewers

To Reviewer #1:

Comment: Several strategies for hindering coke deposition and metal sintering in the DRM have been presented in this review. The manuscript is well-structured, and its content is clearly demonstrated with innovation. I believe that this study is of great relevance to the readers of Catalysts and I recommend its publication after some minor revisions are carried out.

Response: Thank you for your positive comments as well as valuable suggestions to improve our manuscript. Detailed response has been added as below.

Q1: This review is focused on Ni/Al2O3 catalysts. Why have the authors selected this specific catalyst for the DRM? Could the strategies proposed for avoiding metal sintering and coke deposition be similar to other catalysts?

Response: Thank you for your valuable academic comments. Due to the low cost and high surface area of Al2O3, it is a suitable support material especially for industrialization of Ni catalysts for DRM reaction. Moreover, less carbon deposition can be formed on Ni/Al2O3 catalyst due to the stronger MSI between Ni and Al2O3 and the formation of smaller Ni particles. This was related to the formation of NiAl2O4 spinel phase between the Ni precursor and Al2O3 during high temperature calcination. As for the strategies mentioned to promote the performance of Ni/Al2O3, some of them are promising to be applied for other catalysts, such as oxygen vacancies, basicity and MSI.

Q2: The authors have provide a good overview of the most influential features (acidity, specific surface area, metal dispersion…) that affects the performance of Ni/Al2O3 catalyst on the DRM. However, the suitable catalyst design of Ni/Al2O3 catalyst is not specific of DRM, but also is common to other processes such as steam reforming reaction. The authors are suggested to include it in the text.

Response: Thank you for your valuable comments. The contents have been added in the context in red and shown as below:

It is believed that these strategies proven effective in DRM reaction will be promising in modifying the catalysts for other reactions such as steam reforming and tar reforming reactions, thus paving the way for a wide range of catalytic productions of syngas and clean energies.

Q3: In Section 3, the influence of acidity/basicity on the coke deposition is established. However, in this section, the only way presented to modulate Ni/Al2O3 acidity is the incorporation of alkaline metal oxides. The authors are suggested to include how catalyst synthesis and Al2O3 pretreatment may affect catalyst acidity. Thus, it is well-established that the acidity of the Al2O3 support is related to the crystalline phase of the Al2O3 support. Thus, the acid sites of γ-Al2O3 promote carbon deposition on the catalyst surface, which is the main cause of catalyst deactivation in reforming reactions. [https://doi.org/10.1021/acs.energyfuels.1c01666].

Response: Thank you for your valuable suggestions. The revised statements in this part has been updated in the main context in red and shown as below:

Besides the addition of basic oxides, pretreatment atmonsphere can affect the surface acidity and basicity. Compared with air calcination, Ar pretreatment could enhance the basic sites together with the addition of ethylene diamine tetraacetic acid (EDTA) due to the promotional effect of carbon residues at the interface of Ni0 and La2O3 [49]. In another case, N2 calcination would increase the amount of Lewis acid sites on Al2O3, which strongly interacted with metallic Ni and reduced the Ni electron density, thus inhibiting metal sintering and carbon deposition [39,50-52].

Generally speaking, as an acidic support, Al2O3 induces methane cracking and slows down the activation of CO2, thus resulting in carbon deposition. The acidity and basicity of the catalyst can be adjusted by adding alkaline metal oxides such as La2O3 and MgO or non-metal P and modifying the synthesis method such as pretreatment atmosphere to promote the gasification of coke. However, too high alkalinity will also lead to carbon deposition. Therefore, the basicity of the catalyst should be optimized to balance the activity and stability of Ni/Al2O3 in the DRM reaction.

Q4: In order to complete Table 1 with the main points addressed in the review, the authors are suggested to include more studies in which Ni/Al2O3, Ni-M/Al2O3 (being M noble metal, transition metals or Sn) are used in the DRM.

Response: Thank you for your valuable suggestions. More studies have been summarized in Table 1 accordingly.

Q5: In Section 6: Conclusions and prospects, the contents about shortcomings and future development are not adequately covered in conclusion. The authors are suggested to expand this section in terms of the main challenges and more detailed directions for future research.

Response: Thank you for your valuable suggestions. The revised prospects are updated in the main context and shown as below:

First, most of the modifications are limited to one or two functions. Certain improvements have to be at the expense of affecting other properties or performances. For example, addition of Sn could slow down the methane cracking and inhibit the coke formation; however, the coverage on the Ni active sites may reduce the conversions. In this situation, multi-functional catalysts are suggested to be developed to enhance the activity, stability and anti-deactivation properties simultaneously by taking advantages of the synergistic effects of each components in the catalyst. For instance, when mesoporous La2O3 is doped in Ni/Al2O3, the Ni dispersion, surface basicity, MSI and oxygen defects can be adjusted to a suitable level, achieving a robust and active catalyst.

Second, the specific role of NiAl2O4 spinel phase is still in debate that either a promotional effect on the MSI or a negative impact on the metal active sites is reported for DRM reaction. More systematic and mechanism studies are recommended to elucidate the exact formation process and the influences of the resultant spinel structure on the catalytic performance and deactivation behavior.

Third, excessive basicity and MSI may not benefit the performance. Thus, an investigation on the amount of promoters (e.g. MgO) is recommended to conduct, as well as the thermal treatment parameters, in order to achieve a suitable degree of basicity and MSI for an optimized adsorption/activation of CO2 molecules and generation of metal active sites for CH4 dissociation.

Reviewer 2 Report

This manuscript presents interesting review on the dry methane reforming. Different strategies are discussed to enhance the anti-deactivation property of Ni/Al2O3

 I recommend to accept the manuscript with minor revision

  1. Page 3. Line 5 in Section 2. Structure and morphology control

Please revise the following sentence:

«Although Ni/Al2O3 catalyst has a good catalytic activity and stability, it also bears the disadvantages such as coke deposition, sintering, phase transformation and reduction, which deactivates Ni/Al2O3 quickly in the DRM reaction, despite the initial good activity [21-22].»

If Ni/Al2O3 catalyst has a good stability, why it deactivates quickly in the DRM reaction?

The first part of sentence conflict with the second part.

  1. Page 8.

Please check the place of references in the sentence:

Damyanova et al. [58] showed that the strong interaction between Rh and Ni not only improved the reducibility of NiO, but also increased the surface distribution of Ni, producing a small and uniform Ni particle (5 nm in diameter) evenly distributed on the Al2O3 support [59].

It looks like ref [59] is not on the right place.

Author Response

To Reviewer #2:

Comment: This manuscript presents interesting review on the dry methane reforming. Different strategies are discussed to enhance the anti-deactivation property of Ni/Al2O3

 I recommend to accept the manuscript with minor revision

Response: Thank you for your positive comments as well as valuable suggestions to improve our manuscript. Detailed response has been added as below.

Q1: Page 3. Line 5 in Section 2. Structure and morphology control

Please revise the following sentence:

«Although Ni/Al2O3 catalyst has a good catalytic activity and stability, it also bears the disadvantages such as coke deposition, sintering, phase transformation and reduction, which deactivates Ni/Al2O3 quickly in the DRM reaction, despite the initial good activity [21-22].»

If Ni/Al2O3 catalyst has a good stability, why it deactivates quickly in the DRM reaction?

The first part of sentence conflict with the second part.

Response: Thank you for your valuable academic comments. The revised sentences have been updated in the main context and shown as below:

Despite the admirable catalytic performance of Ni/Al2O3 catalyst in methane activation and CO2 conversion, it also bears the disadvantages such as coke deposition, sintering, phase transformation and reduction [21-22].

Q2: Page 8.

Please check the place of references in the sentence:

Damyanova et al. [58] showed that the strong interaction between Rh and Ni not only improved the reducibility of NiO, but also increased the surface distribution of Ni, producing a small and uniform Ni particle (5 nm in diameter) evenly distributed on the Al2O3 support [59].

It looks like ref [59] is not on the right place.

Response: Thank you for your valuable comments. Reference [59] ([62] as updated) has been moved before the reference 58 ([63] as updated) to support the H2 spillover mechanism in the same paragraph.
